

# Aerosol-cloud impacts on aerosol detrainment and rainout in shallow maritime tropical clouds

Gabrielle R. Leung[1], Stephen M. Saleeby[1], G. Alexander Sokolowsky[1], Sean W. Freeman[1], and Susan C. van den Heever[1]

5   [1]Department of Atmospheric Science, Colorado State University, Fort Collins, 80521, USA

*Correspondence to*: Gabrielle R. Leung (gabrielle.leung@colostate.edu)



**Abstract.** This study investigates how aerosol-induced changes to cloud properties subsequently influence the overall aerosol budget through changes to detrainment and rainout. We simulated an idealized field of shallow maritime tropical clouds using the Regional Atmospheric Modeling System (RAMS) and varied the aerosol loading and type between model runs to create a 16-member ensemble. The full aerosol budget was tracked over the course of the 48-hour simulation, showing that increasing the aerosol loading leads to an increase in aerosol regeneration and detrainment aloft at the expense of aerosol removal via rainout. Under increased aerosol loadings, cloud droplets are smaller and more likely to evaporate before they form precipitation-sized hydrometeors. As a result, the aerosol particles contained inside these droplets are released into the environment rather than being removed to the surface via rainout. However, the few raindrops which do happen to form under increased aerosol loadings tend to be larger since the cloud water available for collection is divided among fewer raindrops, and thus raindrops experience less evaporation. Thus, in contrast to previous work, we find increases in aerosol loading lead to decreases in aerosol rainout efficiency even without a decrease in the overall precipitation efficiency. We further used *tobac*, a package for tracking and identifying cloud objects, to identify shifts in the overall cloud population as a function of aerosol loading and type, and found contrasting aerosol effects in shallow cumulus and congestus clouds. Shallow cumulus clouds are more sensitive to the increase in cloud edge/top evaporation with increased aerosol loading, and thereby tend to rain less and remove less aerosol via rainout. On the other hand, larger congestus clouds are more protected from evaporation and are thereby able to benefit from warm-phase invigoration. This leads to an increase in rain rates but not in domain-wide aerosol rainout, as the domain-total rainfall becomes concentrated over a smaller horizontal area. Trends as a function of aerosol loading were remarkably consistent between the different aerosol types tested. These results represent a pathway by which a polluted environment not only has higher aerosol loadings than a pristine one, but is also less able to regulate those loadings by removal processes, instead transporting aerosols to the free troposphere where they remain available for reactivation and further aerosol-cloud interactions.

## 1 Introduction

Clouds play an important role in governing the atmospheric aerosol budget through a number of simultaneous processes that transport, modify, and remove aerosols. One example of such a process occurs when aerosols are transported between the boundary layer and the free troposphere in cloud updrafts and downdrafts (Cotton et al., 1995; Engström et al., 2008; Chen et al., 2012; Twohy et al., 2017; Savre, 2021; Bardakov et al., 2022). Another such process involves aerosol particles being removed from the free atmosphere and undergoing heterogenous chemistry when they are activated as cloud condensation nuclei (CCN) or are otherwise intercepted by hydrometeors (i.e., impaction) (Feingold and Kreidenweis, 2000; Hegg et al., 2004; Ervens et al., 2018). Cloud processes near cloud top and cloud edges may further impact the aerosol budget through the evaporation of hydrometeors, which detrains aerosol particles and impacts aerosol number concentrations and vertical distributions (Herbener et al., 2016; Corr et al., 2016; Leung and van den Heever, 2022). Alternatively in regions where different microphysical processes are dominant, hydrometeors may grow to become precipitation-sized and fall to the surface,



thus removing aerosol particles from the atmosphere via wet deposition (i.e., rainout or washout) (Radke et al., 1980; Kipling et al., 2016).

While clouds transport and remove aerosols, aerosols can in turn influence cloud properties both directly and indirectly. Aerosol particles scatter and/or absorb radiation (direct effect), which can alter cloud development through changes to surface fluxes and atmospheric stability (McCormick and Ludwig, 1967; Atwater, 1970; Kim et al., 2014; Grant and van

den Heever, 2014; Lee et al., 2014; Park and van den Heever, 2022; Sokolowsky et al., 2022). Aerosol particles have also been found to influence the microphysical properties of clouds (indirect effect) with impacts on cloud lifetimes, cloud  types, and the overall cloud fraction (Twomey, 1977; Albrecht, 1989; Tao et al., 2012) as well as precipitation efficiency (Jiang et al., 2010; Dagan et al., 2015).

Assessing the overall change to the aerosol budget for a given perturbation to aerosol-cloud interactions is complex.

Aerosol-induced changes to clouds may feed back to how clouds and precipitation influence the aerosol field, although findings in the literature appear to be mixed. For example, Cui and Carslaw (2006) found that increases in aerosol loading led to decreases in the efficiency of both precipitation and wet scavenging by deep convective clouds. On the other hand, while Lee and Feingold (2010) found similar trends for stratiform clouds, they determined that aerosol loading had only a minor impact on convective precipitation and scavenging efficiencies. Discrepancies such as these are difficult to resolve given the numerous

cloud and aerosol processes involved. Furthermore, differences in simulated environment, aerosol type, and cloud types may also influence aerosol-cloud interactions (Khain et al., 2008; Fan et al., 2009; van den Heever et al., 2011; Altaratz et al., 2014; Grant and van den Heever, 2014; Gryspeerdt et al., 2014; Glassmeier and Lohmann, 2016; Jiang et al., 2018; Dagan and Stier, 2020). Despite these uncertainties, understanding how aerosol-cloud interactions impact the many processes controlling the aerosol budget—as well as which of those impacts are most relevant for a given cloud scene—is essential for representing

realistic aerosol distributions and thus for assessing the ultimate aerosol impacts on weather and climate (Haywood and Boucher, 2000; Samset and Myhre, 2011; Boucher et al., 2013).

In this work, our goal is to examine how aerosol impacts on shallow maritime tropical clouds feed back on the aerosol budget via changes to aerosol rainout and aerosol detrainment. More specifically, we aim to answer the following two

questions: (1) How does the proportion of aerosol particles rained-out versus those detrained aloft change as aerosol loading increases? and (2) How do the changes to the aerosol budget arising from rainout and detrainment vary as a function of aerosol type? We address these questions using a large ensemble of high-resolution simulations of an idealized cloud field under a range of aerosol loadings and types, as described in **Sect. 2**. Trends in domain-wide aerosol budget, cloud properties, and microphysical process rates are presented in **Sect. 3**, while a state-of-the-art cloud tracking package is used to separate trends

among different cloud types in **Sect. 4**. Finally, the role of aerosol type is discussed in greater detail in **Sect. 5**.



## 2 Model description and analysis approach

### 2.1 Model description and configuration

The Regional Atmospheric Modelling System (RAMS; version 6.3.03) is a three-dimensional, non-hydrostatic, cloud-resolving model with a two-moment bin-emulating microphysics scheme (Pielke et al., 1992; Cotton et al., 2003; Saleeby and van den Heever, 2013). Details of the model grid configuration, initial conditions, and parameter settings are described in **Table 1**. The RAMS model configuration used here was identical to that in Leung and van den Heever (2022), though the vertical grid is extended to reach from the surface to ~23 km in altitude. The high horizontal, vertical, and temporal resolution ($\Delta x = 100m$, $\Delta z = 50-300m$, $\Delta t = 0.75s$) allowed the model to resolve large turbulent eddies and represent a wide range of convection over the two diurnal cycles (48 hours) that were simulated. The domain was located entirely over the ocean, and the simulation was initialized using a combination of dropsonde observations from the Cloud, Aerosol, and Monsoon Processes Philippines Experiment (CAMP²Ex) (Reid et al. 2022) and ERA-5 data as described in Leung and van den Heever (2022). After initialization, the model was allowed to evolve freely without additional large-scale forcing. As such, these simulations serve as an idealized representation of the microphysical, dynamical, and radiative processes driving maritime tropical convection.

**Table 1: RAMS model parameters used in the simulation.**

| Model Aspect | Setting |
|---|---|
| Grid | Arakawa C grid |
| | 1000 x 1000 points, $\Delta x = \Delta y = 100m$ |
| | 120 vertical levels, $\Delta z = 50-300\ m$ |
| Time integration | 48-hour simulation duration, $\Delta t = 0.75s$ |
| Initialization | Horizontally homogenous thermodynamic and wind profile, averaged from ERA-5 and CAMP²Ex dropsonde |
| | Random potential temperature perturbations within the lowest 500m AGL of the domain, with a maximum perturbation of 0.1K |
| Surface scheme | All-ocean surface with spatially- and temporally-uniform sea surface temperature (SST=29ºC) |
| | LEAF-3 (Walko et al., 2000) |
| Boundary conditions | Periodic in zonal and meridional directions |
| Microphysics scheme | Two-moment bulk microphysics (Meyers et al., 1997) |
| | 8 hydrometeor classes (Saleeby and Cotton, 2004) |
| | Heterogenous ice nucleation (DeMott et al., 2010) |
| Radiation scheme | Two-stream, hydrometeor sensitive (Harrington, 1997) |
| | Updated every 5 minutes |





| Aerosol treatment | Maximum concentration at the surface and exponentially decreasing with altitude with scale height of 7km |
| --- | --- |
| | Aerosol-radiation interactions on |
| | Aerosol sources and sinks on, with full aerosol budget tracking (Saleeby and van den Heever 2013) |

The aerosol field was initialized homogenously in the horizontal direction at the first model timestep but decays

exponentially in the vertical direction with a scale height of 7 km. Both aerosol-radiation and microphysics-radiation interactions were included in the simulation. The aerosol budget capabilities in RAMS allowed for the tracking of aerosol number and mass in the following categories: (1) unactivated, (2) in-hydrometeor, (3) regenerated, and (4) wet-deposited/rained-out aerosol (Saleeby and van den Heever, 2013). A schematic depicting the processes governing the exchange between these categories is shown in **Figure 1**. Aerosol particles were initialized in the unactivated aerosol category. Over

time, the aerosol number and mass concentration fields changed freely as particles were advected around the domain. If the aerosol particles were entrained into an updraft and encountered sufficient supersaturations to activate and serve as CCN, they were transferred to the in-cloud aerosol category. As water mass was transferred between hydrometeor species (i.e., cloud, drizzle, rain, ice, snow, aggregates, hail, graupel), a corresponding fraction of aerosol was also transferred. Under subsaturated conditions, the hydrometeors evaporated and the aerosol particles acting as CCN/INP were returned to the environment as

regenerated aerosol. Finally, if aerosol particles were contained within raindrops which fell to the ground (either because they were activated as CCN or because they were intercepted by a hydrometeor), they were transferred to a category tracking the accumulated aerosol mass reaching the surface via wet deposition or rainout.





**Figure 1: Schematic of processes involving aerosol transfer represented in the ensemble of RAMS simulations. The aerosol budget**
**terms described in text are depicted in gray boxes. Black arrows depict the transfer of aerosol number and mass between the different**
**budget terms.**

## 2.2 Experiment set-up

The full model ensemble presented here consists of sixteen simulations in which four aerosol loadings and four aerosol types

were varied (**Table 2**). The four aerosol loadings tested span the observed range of aerosol loadings during the CAMP[2]Ex field

campaign, from clean to highly polluted environments (Reid et al., 2022). Throughout the rest of this paper, the different

aerosol loading runs are denoted by the initial aerosol number concentration at the surface (which was also the maximum





aerosol number concentration initialized in each column), which are 100, 500, 1000, and 1500 cm$^{-3}$. The four aerosol types tested were ammonium sulphate, sea salt, mineral dust, and absorbing carbon. These aerosol types have varying median particle

sizes, solubility and hygroscopicity, and radiative properties. The aerosol size distribution was represented as a single log-normal mode with a shape factor of 1.8 (Reid et al., 2022), with the median particle size depending on the aerosol type as specified in **Table 2**. Other key properties of the different aerosol types are also listed in **Table 2**.

**Table 2: Key aerosol parameters for different aerosol types in RAMS. Radiative parameters are given for RH 80%, size bin 14 (particle diameter 0.16 μm), radiation band 3 (visible).**

| Aerosol Type | Median Particle Diameter (μm) | Solubility Fraction | Density (kg m$^{-3}$) | Hygroscopicity | Qext | Qscat | SSA |
|---|---|---|---|---|---|---|---|
| Ammonium Sulphate | 0.18 | 0.9 | 1857.1 | 0.651 | 2.07630 | 2.01450 | 0.97 |
| Sea Salt | 0.2 | 1 | 2165 | 1.334 | 2.06070 | 2.06070 | 1 |
| Absorbing Carbon | 0.1 | 0.05 | 2605.95 | 0.053 | 2.08970 | 1.13680 | 0.544 |
| Mineral Dust | 0.1 | 0.05 | 2463.45 | 0.050 | 2.09050 | 1.43280 | 0.685 |

**2.3 Cloud identification and tracking**

Individual cloud updrafts in each simulation were identified and tracked using the Tracking and Object-Based

Analysis of Clouds (*tobac*; version 1.5.0 release candidate 1) package (Heikenfeld et al., 2019; Sokolowsky et al., 2022). We use the term "feature" to refer to an updraft region at a given timestep, and "cell" to refer to a given cloud feature tracked across time. First, updraft features were identified at each analysis timestep (at a frequency of 5 minutes) based on the three-dimensional vertical velocity field. Regions of local maximum vertical velocity were identified as features at three threshold values (1, 3, and 5 m s$^{-1}$), and a centroid position was assigned to each feature. Secondly, features in subsequent timesteps

were linked based on their inferred motion to create a cell with a trajectory over time. Any cells which had a lifetime of less than 5 minutes (i.e. cells which were tracked for only a single analysis timestep) were excluded from the analysis as part of the quality control. Finally, a contiguous three-dimensional cloudy region (where cloud condensate was above 0.01 g kg$^{-1}$) was identified around each updraft feature using the *tobac* watershed segmentation technique. Cloud top and base altitudes, as well as cloud areas and volumes, were calculated based on the size of this cloudy region. Similar segmentation was performed

on the two-dimensional surface precipitation and aerosol rainout rates in order to identify the size of the raining/rainout area associated with each cloud. Based on these identified features and cells, the mean and maximum values for variables such as



rain rate, aerosol rainout rate, and updraft velocity were calculated for each cloud over a given timestep and over its entire lifetime. Other studies have used previous versions of *tobac* to effectively track cloud objects in a similar manner (e.g. Marinescu et al., 2021), but recent improvements to *tobac* (Sokolowsky et al., 2022) have specifically allowed for cloud objects

to be identified and tracked in three-dimensions and across periodic boundaries as was necessary for this study.

It should be noted that in this work, we are tracking on updrafts, meaning that clouds at the very beginning or end of their life cycles with updrafts weaker than 1 m s$^{-1}$ are necessarily excluded from the analysis. This is a limitation of tracking packages that use a physically-informed threshold to detect features; however, since *tobac* allows for setting multiple thresholds, we have set a fairly low minimum threshold in order to capture the majority of the cloud lifetime. We found that a

vast majority of the falling rain and rained out aerosol (>80%) could be attributed to the *tobac*-tracked cloud features meeting our QC thresholds, and that this was consistent across all these simulations.

## 3 Aerosol impacts on domain properties

In this section, we examine the differences in domain-wide properties as a function of aerosol loading and type. Qualitatively similar cloud fields develop in all sixteen simulations, consisting primarily of shallow cumulus (with cloud tops ~2-4 km AGL)

and congestus (with cloud tops ~4-7 km AGL). Clouds begin to form in the simulations after 6-7 hours and the cloud field develops a variety of cloud morphology and degrees of organization over the next 48 hours (e.g. linear groups, scattered/isolated clouds, arc clouds associated with cold pools). Deep convection (with cloud tops >7 km AGL) occurs only sporadically in a handful of the simulations and does not persist in any of them. An example of the cloud scenes typically simulated is shown in **Figure 2**.

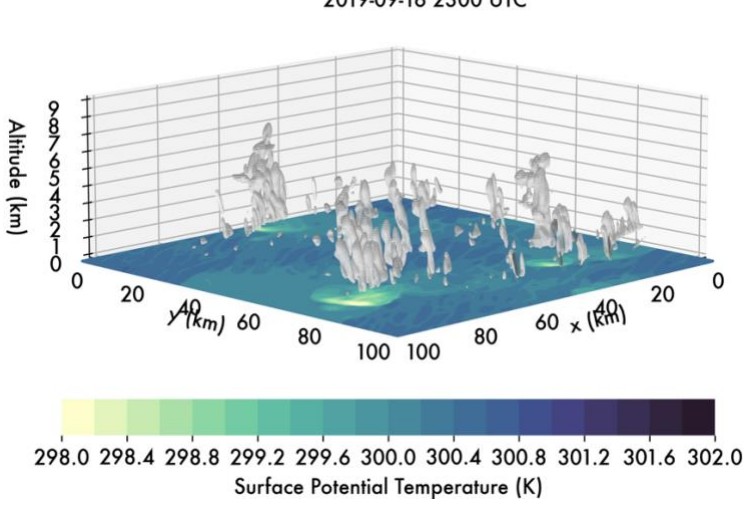


**Figure 2: Three-dimensional rendering of simulated cloud field at 23:00UTC (7:00 LT) for the control sea salt simulation (initial number concentration of 100 cm$^{-3}$ at the surface). Gray isosurfaces are 0.01 g kg-1 of cloud condensate. Surface colours are the potential temperature at the lowest model level above the surface, with lighter colours highlighting the development of cold pools associated with the clouds.**






### 3.1 Domain-wide aerosol budget

The domain-wide trends in the aerosol mass budget are presented in **Figure 3**. For each simulation, we integrate the amount of aerosol mass in each of the four budget categories (namely unactivated, in-hydrometeor, regenerated, and rained-out) after 48 hours of simulation time, and then normalize this mass by the total initial mass at the beginning of the model run. In this

manner, we quantify the *percentage* of aerosol mass that is apportioned to each budget category, thus providing a fairer comparison between simulations of different initial aerosol loading. If changes to the aerosol loading have no impacts on the cloud field and cloud processes impacting aerosol particles, then we would expect the same distribution of aerosol mass across the budget categories, irrespective of aerosol loading. We have chosen here to present an aerosol mass budget rather than a number budget, since aerosol mass is conserved after activation and subsequent regeneration of aerosol particles, and therefore

allows us to specifically account for all the transfers between aerosol budget categories. Over the course of the simulation, we found that less than 5% of the aerosol mass is not tracked and is treated as a residual that is lost due to dry deposition and/or numerical diffusion; this residual proportion is very similar across all ensemble members.

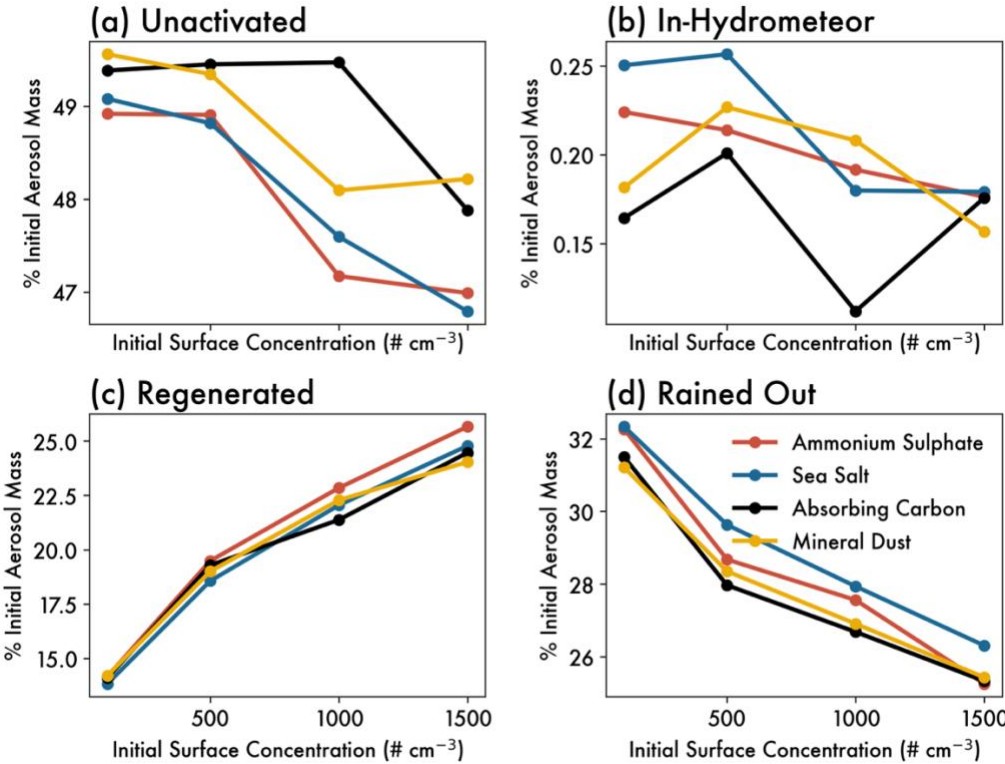

**Figure 3: Domain aerosol mass budget after 48 hours of simulation time represented as a function of the initial aerosol surface**
**concentration (# cm⁻³) and aerosol type. Each aerosol budget term is the domain-integrated aerosol mass in a given category normalized by the total aerosol mass at initialization time, shown for (a) unactivated, (b) in-hydrometeor, (c) regenerated, and (d) rained-out aerosol. See the text for explanations of each aerosol category.**



Across all the ensemble members about half of the initial aerosol mass serves as CCN at some point in time, while
the other half remains unactivated (**Figure 3a**) for the duration of the simulation. As aerosol loading increases, the fraction of
unactivated aerosol generally decreases (the fraction of activated aerosol increases), though changes are fairly small compared
to the trends in the other aerosol budget categories. At first glance, this appears to contradict classic cloud parcel theory,
wherein the activated fraction decreases (unactivated fraction increases) as aerosol number concentration goes up for a given
updraft speed (Reutter et al., 2009). In this scenario, the increased CCN number concentrations lead to increases in cloud
droplets competing to consume the available supersaturation. Thus, that parcel's maximum supersaturation is lower compared
to a parcel with fewer aerosol particles and an identical updraft speed, and it is therefore unable to activate the smallest aerosol
particles, which subsequently drives down the activated aerosol fraction. However, it is important to note that the results we
present in **Figure 3** are not from a single cloud parcel, but rather integrated over the whole cloud field, and therefore incorporate
any aerosol effects that lead to changes in the updraft speed, ambient relative humidity, and cloud types. We discuss these
changes to the broader cloud population in greater detail in **Sect. 4**. There is also a slight decrease in the aerosol mass found
inside cloud droplets or embedded in drizzle or raindrops (**Figure 3b**), though the trend is non-monotonic and inconsistent
between aerosol types. However, the in-hydrometeor category comprises a relatively small percent of the overall aerosol
budget at any given timestep, since aerosol particles are only apportioned to it temporarily before being transferred to the
regenerated or rained-out category.

The clearest trends in the domain-wide aerosol budget as a function of aerosol loading are seen in the proportion of
aerosol mass that is regenerated (**Figure 3c**) or rained-out (**Figure 3d**). It is clear from these figures that the proportion of
regenerated aerosol mass is enhanced while the proportion of rained-out aerosol mass is decreased with increasing aerosol
loading. The opposing trends between the increases in regenerated aerosol and decreases in rained-out aerosol with increasing
aerosol loading have similar magnitudes, and are on the order of 7-10% of the initial aerosol mass. These trends are remarkably
consistent with aerosol type, as will be discussed further in **Sect. 5**. We emphasize here that the aerosol mass budgets we
present are normalized by the initial aerosol loading—in the highest-aerosol loading case, there is not only a greater absolute
mass of aerosol that is regenerated and detrained back into the environment, but also actually a larger percentage of mass
involved. The changes to rainout, which is the only major aerosol sink in the budget examined here, suggests a positive
feedback mechanism by which increases in aerosol loading may actually inhibit the removal of aerosol particles by clouds.

**3.2 Cloud and rain microphysics**

To investigate the mechanisms by which increased aerosol loading leads to aerosol regeneration being favoured over rainout,
we examine trends in the cloud and rain droplet size distributions (**Figure 4**). For all aerosol types, increasing the aerosol
loading produces more numerous and smaller cloud droplets (**Figure 4a-b**), which is consistent with the first indirect effect
that has been demonstrated in many observational and modelling studies (Twomey, 1977; Tao et al., 2012). On the other hand,
increasing aerosol loading produces fewer and larger raindrops (**Figure 4c-d**). This effect has been demonstrated in multiple





modelling studies (Berg et al., 2008; Li et al., 2013; Altaratz et al., 2014; Sheffield et al., 2015) with more limited observational support (Berg et al., 2008; May et al., 2011).

**Figure 4: In-cloud mean microphysical size distribution properties as a function of initial aerosol surface concentration (# cm⁻³) and aerosol type: (a) cloud droplet number concentration, (b) cloud droplet diameter, (c) rain drop number concentration, and (d) rain drop diameter. Values are spatially and temporally averaged over cloudy (cloud condensate > 0.01 g kg⁻¹) updraft (vertical velocity > 1 m s⁻¹) grid points.**


These trends in the droplet size distributions can be more easily connected to the aerosol budget by examining trends

in the microphysical process budget, as shown in **Figure 5**. Each term in this figure represents a different sink for cloud water and is given as an efficiency relative to the mass of water vapor transferred to liquid in cloud droplets. The value given is the percent of condensed water vapor that ends up in a particular cloud water sink. As aerosol loading increases, the cloud droplets are smaller and thus evaporate more readily (**Figure 5a**) and are also less likely to be collected into rain water (**Figure 5b**). This explains the trends in the aerosol budget under increased aerosol loadings to first order: although a similar proportional





mass of aerosol particles is activated and enters cloud droplets, those cloud droplets are smaller and tend to evaporate more quickly, which favors regeneration. Simultaneously, those smaller cloud droplets are less likely to be collected as rainwater, and thus the CCN contained within them are less likely to be washed out within precipitation.

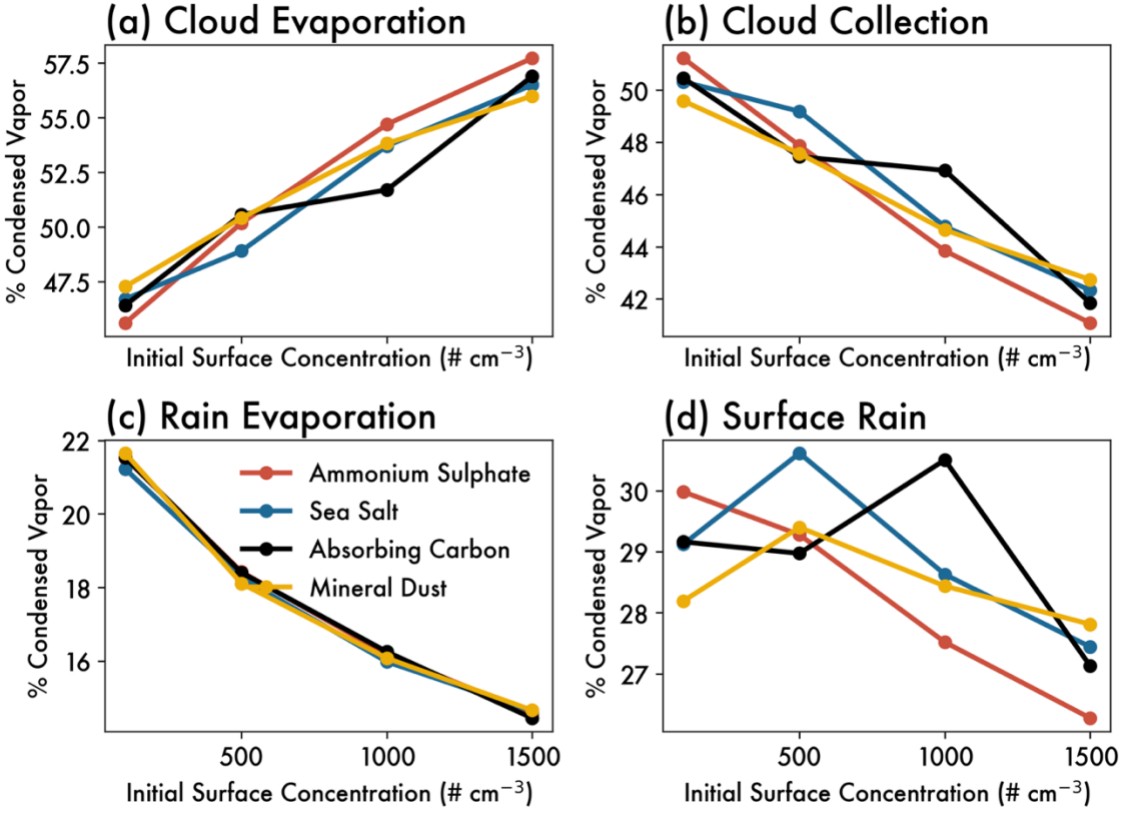

**Figure 5: Domain total microphysical process rates as a function of initial aerosol surface concentration (# cm$^{-3}$) and aerosol type. Each panel shows the process efficiency, defined as the domain- and time-integrated process rate normalized by the domain- and time-integrated condensation rate (water vapor to liquid water), shown for (a) cloud water evaporation, (b) cloud water collection into rain, (c) rain water evaporation, and (d) surface-accumulated rain (i.e. precipitation efficiency).**

In the polluted aerosol environment, although cloud water is less likely to be collected by rain, the cloud water which *is* collected is distributed among fewer raindrops. These raindrops are thus larger in size (**Figure 4d**) and have less surface area compared to raindrops forming in pristine aerosol environments, in keeping with past model results (Storer and Heever, 2013; Altaratz et al., 2014; Saleeby et al., 2015). As a result, even though increasing aerosol loading leads to less efficient collection of cloud water into rainwater, it also leads to decreases in rainwater evaporation (**Figure 5c**). These two trends

partially offset one another, leading to mixed trends in the percentage of cloud water that falls to the surface as rain (**Figure 5d**), sometimes defined as the precipitation efficiency (Cui and Carslaw, 2006; Jiang et al., 2010; Lee and Feingold, 2010).





These decreases with increasing aerosol loading are not monotonic for all aerosol types, and it is a smaller relative trend compared to those of cloud evaporation and collection. Although increasing the aerosol loading in the domain causes clouds to become less efficient at removing aerosol via rainout (**Figure 3d**), it does so without necessarily impacting the precipitation

efficiency itself (**Figure 5d**). Earlier results by Cui and Carslaw (2006) and Lee and Feingold (2010) showed similar decreases in the aerosol rainout efficiency (or aerosol precipitation efficiency or scavenging efficiency, as they respectively referred to it), but these decreases were closely coupled with decreases in precipitation efficiency. That is to say, they showed that increased aerosol loading caused less efficient rain formation and therefore less efficient aerosol removal through rainout. Our results add to this and show that the reduced efficiency in aerosol removal through rainout can occur even without changes to

the domain-wide precipitation efficiency.

## 4 Aerosol impacts on cloud population distributions

### 4.1 Trends in cloud numbers and median cloud properties

Given the complexity surrounding the number of processes impacting rain formation, which often have opposing trends as a function of aerosol loading, we find that increasing aerosol loading does not necessarily lead to decreases in precipitation

efficiency (**Figure 5d**) or the domain-wide accumulated precipitation (**Figure 6a**). For some aerosol types (e.g., mineral dust), the increase in aerosol loading actually leads to an increase in the accumulated precipitation between the lowest and highest aerosol loadings tested in our ensemble. The trends in the total number of tracked clouds are similarly non-monotonic and mixed, as shown in **Figure 6b** and suggest that aerosol loading does not have a clear impact on domain cloudiness for this ensemble of maritime tropical clouds. That being said, tracking the clouds over their full lifetime with the use of *tobac* allows

us to further subdivide clouds into those that precipitate and those which do not (**Figure 6c-d**). To differentiate raining and non-raining clouds, we find the area-mean rain rate associated with each cloud feature at each point in its lifetime, and then take the lifetime-maximum of those rain rates such that non-raining clouds are those which never reach an area-mean rain rate of 0.0001 mm hr$^{-1}$. Clearly, the mixed trend in the total number of clouds as a result of increasing aerosol loading arises as a result of two opposing trends: increases in the number of non-precipitating clouds and decreases in the number of precipitating

ones. This is further consistent with the general picture of aerosol impacts on the cloud field from the process rates, as described above: although clouds still do form in environments with higher aerosol loadings, the cloud droplets evaporate more readily before the cloud is able to produce precipitation-size particles, and thus a greater fraction of clouds never rain throughout their whole lifetime. Non-raining clouds still activate aerosol particles and regenerate them aloft as the cloud dissipates, but they do not remove them from the domain via rainout, which contributes to aerosol regeneration at the expense of rainout.





**Figure 6: (a) Total accumulated rain, and (b-d) number of updraft cells tracked over 48 hours of simulation using *tobac* as a function of initial aerosol surface concentration (# cm⁻³) and aerosol type. Total numbers of tracked cloud cells are shown in (b), while (c) shows only non-raining updrafts and (d) shows only raining updrafts. Raining updrafts are defined as those which have a mean rain rate of at least 0.0001 mm/hr for any timestep during their lifetime.**


      We can use *tobac* to aggregate the raining clouds (**Figure 7**) to see how the median properties of these clouds evolve under different aerosol environments. Although increasing aerosol loading leads to fewer raining clouds, those which *do* rain are invigorated with higher rain rates (**Figure 7a**). This is further compounded by the decrease in the median area covered by each raining cloud (**Figure 7b**). Overall, we find that under higher aerosol loadings, surface rainfall becomes more concentrated amongst fewer clouds with a higher median rain rate. However, this is contrasted by the trends in aerosol rainout rate (**Figure 7c**). The aerosol impact is non-monotonic, but there is a decrease in the strength of aerosol rainout between the lowest and highest aerosol loadings for all aerosol types. We conclude that *the median cloud rains more under higher aerosol*





*loadings but rains out less aerosol.* Shifts in the behaviour of the overall cloud population as well as potential explanations for this behaviour are discussed in the following subsection.


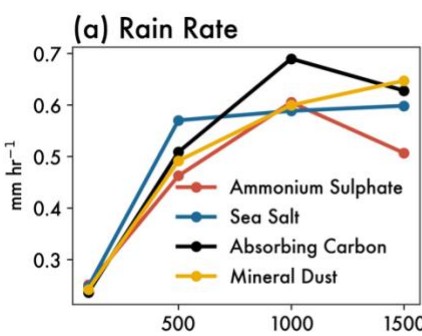 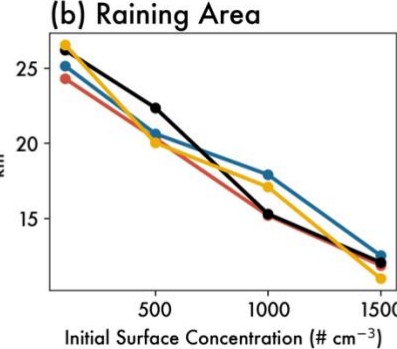 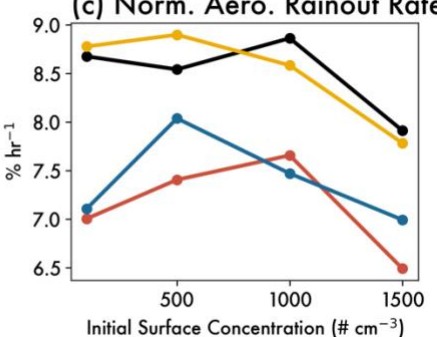

**Figure 7: Median updraft properties for raining updrafts as a function of initial aerosol surface concentration (# cm$^{-3}$) and aerosol type. Panels show (a) rain rate (mm hr$^{-1}$), (b) raining area (km$^2$), and (c) normalized wet deposition rate (% hr$^{-1}$). The normalized wet deposition rate is the percent of initial aerosol mass integrated over a given column that is lost to rainout over a given time**
**period.**

## 4.2 Shifts in cloud population distributions

An advantage of cell tracking over prior analysis methods is the ability to examine trends and characteristics of evolving cloud population distributions as opposed to merely aggregating properties. We construct two-dimensional histograms according to
cloud top height (CTH) and either rain rate (**Figure 8**) or normalized aerosol rainout rate (**Figure 9**), such that the value in each bin is the number of raining clouds with a given CTH and rain rate/rainout rate. The CTH, rain rate, and normalized aerosol rainout rate are defined by taking the lifetime-maximum value for each tracked cloud cell, such that they represent the "peak" maturity for a given cloud. This approach has the benefit of being able to separately identify cloud modes according to different CTHs; in this case, the shallow cumulus mode is clearly visible as a hotspot of clouds with CTHs of ~2 km, as is the
congestus mode with CTHs of ~4 km (e.g., **Figure 8a**) This is particularly important due to our focus on tropical convection, which is known to consist of three separate cloud modes (Johnson et al., 1999; Posselt et al., 2008).





**Figure 8: Two-dimensional histogram of cloud top height (km) and rain rain rate (mm hr⁻¹) for all tracked raining clouds. Each row is a different aerosol type: (a-d) sea salt, (e-h) ammonium sulphate, (i-l) absorbing carbon, and (m-p) mineral dust. The leftmost column (a,e,i,m) is the control run with the lowest aerosol loading, and coloured contours indicate the number of raining cloud cells in each joint probability bin. The other three columns are given as a difference in number of cloud cells relative to the control run of the same aerosol type. The overlaid gray contours correspond to n=250, 300, 350 in the respective control runs, and are drawn to facilitate comparison between the different simulations.**






Generally, there is a positive correlation between CTH and precipitation rate, with taller clouds tending to have stronger rain rates than shallower clouds do (Adler and Mack, 1984; Smalley and Rapp, 2020). Moving from left to right in each row in **Figure 8** allows one to see the impact of increasing aerosol loadings on the CTH-rain rate distribution, such that negative values (blue regions) are portions of the cloud population that become less frequent with increasing aerosol loading, and vice-versa for positive values (red regions). Notably, increased aerosol loadings impact shallow cumulus and congestus

clouds in opposite ways, emphasizing that median cloud properties over all types of clouds are insufficient for quantifying the magnitude of the aerosol effect. Shallow cumulus clouds tend to grow taller, with the modal CTH being closer to 3 km in the highest aerosol simulations, but they do so with reduced rain rates (e.g., **Figure 8d**), in a manner that is similar to the precipitation suppression effect described in many past studies (e.g. Xue et al., 2008; Spill et al., 2019). On the other hand, congestus clouds also grow slightly taller, but have much stronger modal rain rates, which is consistent with previous findings on the warm-phase invigoration of this tropical cloud mode (Li et al., 2013; Sheffield et al., 2015). These results suggest that

the smaller shallow cumulus clouds are more sensitive to the increase in evaporation (**Figure 5a**), whereas congestus clouds with larger areas are more able to protect the interior of the cloud core from evaporation thereby benefitting from the increase in latent heating or warm-phase invigoration associated with increased cloud droplet formation (**Figure 4a**).



**Figure 9: As in Figure 8, but for normalized aerosol rainout rate. The normalized aerosol rainout rate is the percent of initial aerosol mass integrated over a given column that is lost to rainout over a given time period. The overlaid gray contours in the three rightmost columns are n=300,400,500 in the respective control runs.**




The CTH-aerosol rainout rate distribution in **Figure 9** clearly shows deeper clouds being associated with more aerosol
rainout compared to shallower clouds. This relationship is consistent with trends in rain rate with cloud top height as discussed
previously. For shallow cumulus, increased aerosol loading leads to a decrease in their ability to remove aerosol via rainout,
which follows closely with the decrease in their rain rates. However, although congestus clouds tend to produce stronger
rainfall in higher aerosol environments, they do not see a corresponding increase in their ability to rainout aerosol, which has
no change or even decreases slightly for the highest aerosol loadings. These results suggest that unlike rain rates, which can
be enhanced by warm-phase invigoration, there is something of a "saturation effect" for rainout. At a certain point, stronger
rain rates can no longer increase the amount of aerosol being rained-out since the aerosol available to rainout in those areas
has already been removed to the surface. Because the surface rainfall from these clouds becomes concentrated over smaller
horizontal areas where the cloud droplets are sufficiently protected from the environment such that they can form precipitation-
sized hydrometeors, there are increasing areas of clear-sky or very light precipitation that is not sufficient to remove aerosol
particles to the surface. Thus, the overall aerosol impact on rainout is dominated by the decrease in rainout from shallow clouds
which either have weaker rain rates or stop raining altogether. These findings provide strong process-level evidence for a
potential mechanism to explain recent results from GCMs showing that frequent, light precipitation is more important than
strong precipitation in regulating the amount of wet deposited aerosol (Wang et al., 2021a, b).

## 5 Influence of aerosol type

We have shown throughout this paper that the influence of aerosol type on the overall aerosol budget and cloud populations is
relatively small. Regardless of aerosol type, increasing aerosol loading leads to similar trends in cloud microphysics,
precipitation rates, and the domain-wide aerosol budget and vary only in terms of magnitude. The different aerosol median
sizes and hygroscopicities (**Table 2**) do influence the magnitude of the aerosol rainout, although only minimally, and they do
not affect the overall trends. For example, the cloud field overall tends to be more efficient at raining out ammonium sulphate
and sea salt (**Figure 3d**), both of which have larger particle sizes, which is consistent with these particles having a higher
activation fraction all else being equal (Reutter et al., 2009). Note that comparing among the median cloud in each simulation
(**Figure 7c**) shows lower normalized aerosol rainout rates for ammonium sulphate and sea salt. This is driven by the tail of the
distribution shown in **Figure 9**—since the clouds are more efficient at raining out ammonium sulphate and sea salt, cumulus
clouds with low rainout rates are still able to show appreciable and trackable values. However, the integrated impact of all
clouds in the field is more accurately shown in **Figure 3d** or by looking at the full distribution in **Figure 9**, which emphasizes
the importance of evaluating changes across the whole cloud distribution and not merely in medians or means across different
cloud modes.

We found that cloud properties depend more strongly on the magnitude of aerosol loading rather than the aerosol
type. The lack of variation in the cloud population distribution as a function of aerosol type (as can be observed by comparing




across each column in **Figure 8** and **8**) was observed despite the strong differences in clear-sky radiative heating rates as a function of aerosol type (**Figure 10**). The differences in heating rates are driven by differences in the absorbing/scattering properties of each aerosol type, with more scattering aerosol, like ammonium sulphate or sea salt, driving cooling trends with increased aerosol, and more absorbing aerosol, like absorbing carbon and mineral dust, driving warming and stronger stratification of the stable layers in the domain. However, these differences in radiative heating do not appear to feed back on cloud properties within the domain—at least not on the timescales of our simulation (48 hours) — as there is not sufficient time for these differences in direct aerosol effects to influence the overall cloud field. These differences may eventually lead to divergence between aerosol types as the system moves towards radiative-convective equilibrium (RCE), though there is some debate about whether the relatively short lifetimes of these shallow cumulus and congestus clouds allows for such equilibration, and longer-term and realistically-forced simulations would be necessary to test this (Dagan et al., 2018). Some past research also suggests this may be sensitive to the update timescale used for the radiation parameterization (Matsui et al., 2020). We would also expect that the radiative differences between aerosol types would be amplified over a land surface where aerosol-induced differences in the feedbacks to the surface fluxes could play a role, in contrast to the ocean surface (here, we have fixed the SST, though we would not expect large aerosol-induced changes in SST over the 48 hours of simulation time even if a fully interactive ocean surface were utilized).

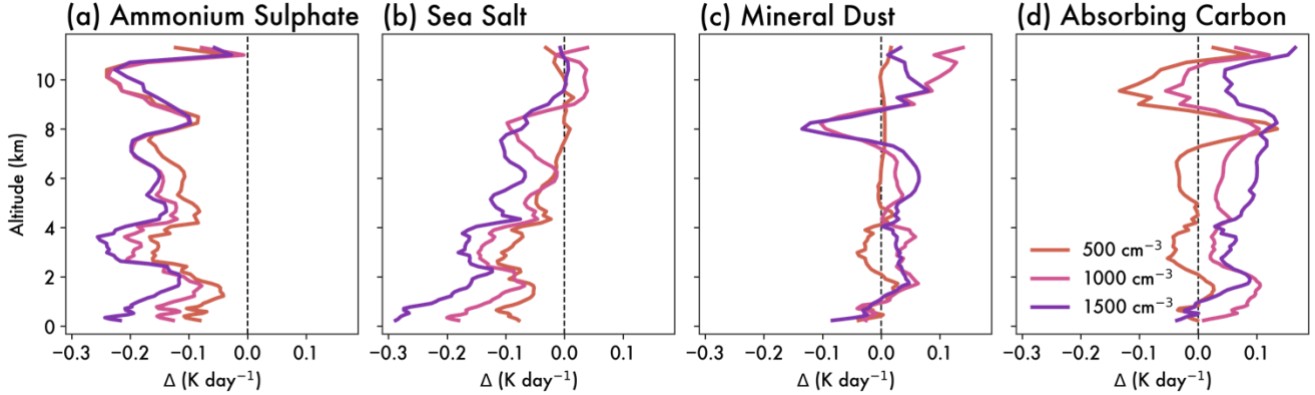

**Figure 10: Vertical profile of domain-mean clear-sky radiative heating rates for different aerosol types: (a) ammonium sulphate, (b) sea salt, (c) mineral dust, and (d) absorbing carbon. Coloured lines indicate different initial aerosol concentrations. All profiles are given as a different relative to the lowest-aerosol loading simulation (initial concentration of 100 # cm$^{-3}$ at the surface).**

## 6 Summary and discussion

Aerosol-cloud interactions are a key uncertainty both on the overall climate forcing and on weather timescales. They are especially challenging to unravel given the large number of processes and feedbacks involved. Here, we have aimed to examine how aerosol impacts on shallow maritime tropical clouds (cumulus, congestus) feed back to the aerosol budget, specifically through the removal of aerosol via rainout versus the regeneration of aerosol particles via detrainment aloft. By combining a suite of high-resolution simulations with a comprehensive apportionment of an aerosol budget facilitated by the *tobac* tracking



package, we were able to track both domain-wide trends and the processes driving those trends in different modes of the tropical cloud population as a function of aerosol loading and aerosol type.

First, we examined trends in the domain-wide aerosol budget and cloud microphysical processes. We found that regardless of aerosol type, increasing the aerosol loading enhances aerosol regeneration at the expense of rainout. This effectively hinders the cloud field's ability to remove aerosol, and thus represents a positive feedback by which increased

aerosol loadings may strengthen with time. The increased aerosol regeneration is driven by a decrease in the size of cloud droplets and thus an increase in evaporation. Cloud droplets are therefore increasingly likely to evaporate before the cloud has the chance to form rain and hence remove aerosol through surface deposition. Although there are fewer raindrops that form, those which do form through stochastic collisions with other droplets then each have more cloud water available for collection, since a similar amount of cloud water is being divided among fewer raindrops. As a result, these raindrops can be larger and

thus experience less evaporation. Previous work has shown increasing aerosol loadings to be associated with decreases in precipitation efficiency and aerosol rainout efficiency (Cui and Carslaw, 2006; Lee and Feingold, 2010)—here, we show that the decrease in the cloud field's efficiency at removing aerosol via rainout can occur even without a corresponding decrease in the overall precipitation efficiency. In other words, even when clouds produce a similar amount of precipitation under higher aerosol loadings due to compensating changes to cloud and rain microphysics, the ability of clouds to remove aerosol via

rainout is still hampered because it depends primarily on changes to cloud water collection.

Additionally, we used *tobac* to track clouds over the course of their lifetime and generate statistics of cloud properties. We found that increases in aerosol loading lead to more non-raining clouds and fewer raining clouds; this again is in keeping with the idea that precipitation is suppressed and clouds tend to dissipate before they can form rain. Those clouds which do rain tend to have stronger rain rates over a smaller horizontal area, such that precipitation is increasingly concentrated among

fewer clouds that are warm phase invigorated under high aerosol conditions. Although these clouds then have stronger rain rates, this does not lead to more aerosol rainout overall, due to the smaller horizontal area covered by rain leaving a larger clear-sky region where aerosol particles can remain in the atmosphere. This further validates our initial domain-wide analysis showing decreases in aerosol rainout efficiency without changes to precipitation efficiency.

Finally, we examined changes caused by varying aerosol environments as a function of different cloud types and

found that a mixed domain-wide trend in rain amounts that is driven by contrasting aerosol effects in shallow cumulus and congestus clouds. With increasing aerosol loading, shallow cumulus clouds tend to grow taller and rain less or not at all, whereas congestus clouds only grow slightly taller and tend to have higher rain rates. These changes in rain intensity lead to shallow cumulus clouds being less able to remove aerosol via rainout. Despite congestus having higher rain rates with increased aerosol, there is a "saturation effect" such that the more intense rain no longer increases the efficiency with which aerosol is

rained-out, since the rainfall depletes most of the available aerosol even before the rain rates intensify. As a result, the decreases in shallow cumulus precipitation dominate the aerosol effect on the overall amount of aerosol which is removed by rainout. Furthermore, these results underscore that aerosol-cloud interactions can be highly dependent on cloud type, given that the



balance between susceptibility to evaporation and warm-phase invigoration depends on the cloud size and dominant microphysical processes, and ultimately determines the sign of the precipitation response.

In general, we found that the magnitude of aerosol loading tested in our ensemble had a stronger impact on aerosol-cloud impacts than did the aerosol type, despite the clear differences in the radiative heating rates brought about by the latter. We suggest that the differences in the aerosol direct effect between different aerosol types may pose a stronger impact over longer timescales and/or over land surfaces that have more rapid surface flux feedbacks, and recommend future work be undertaken to investigate these scenarios. We also caution that while these results are robust over this suite of simulations with

varied aerosol environments, the simulated meteorology only captures a particular maritime tropical environment, and the strength of the aerosol budget response may depend on other factors including the large-scale meteorology. However, the consistent trends in aerosol impacts on the microphysical processes, the whole cloud population distribution and the domain aerosol budget, suggest that the aerosol-cloud interactions described here may be significant. These interactions represent a pathway by which *a polluted environment not only has higher aerosol loadings than a pristine one, but is actually less able to*

*regulate those loadings by removing aerosol—instead, the aerosol is convectively transported from the boundary layer to the free troposphere*, where aerosol particles remain available for reactivation and further aerosol-cloud interactions.

**Data and code availability**

The source code for RAMS, namelist files, and other information and analysis scripts necessary to reproduce the simulations will be made available through: https://github.com/grleung/aerobudget. Source code for the tobac package is available through:

https://github.com/tobac-project/tobac.

**Author contributions**

GRL and SCvdH designed the experiments. GRL conducted the RAMS simulations. SMS made developments to RAMS to facilitate the analysis presented here. SWF and GAS made developments to and assisted in the implementation of *tobac* for the cloud identification and tracking used in the analysis. GRL and SCvdH performed the data analysis and prepared the paper,

with contributions and edits from all co-authors.

**Competing interests**

The authors declare that they have no conflict of interest.

**Acknowledgements**

This research was supported by NASA CAMP²Ex Grant 80NSSC18K0149.





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
