# Peer review of "Aerosol-cloud impacts on aerosol detrainment and rainout in shallow maritime tropical clouds"

_EGUsphere, 2022_

## Referee Comment (RC2)

The authors have investigated aerosol-induced changes to cloud properties in shallow maritime tropical clouds. The modelling study demonstrates that increasing the aerosol loading enhances aerosol regeneration at the expense of rainout regardless of aerosol type. Such studies are very important for a process-based understanding of aerosol-cloud interactions. However, clarity and precision are lacking in the manuscript. However, clarity and precision are lacking in the manuscript.

The topic "Aerosol-cloud impacts on aerosol detrainment and rainout in shallow maritime tropical clouds" is of interest and fits the scope of EGUsphere. The paper can be accepted with major revisions. Please see the comments below.

**Major Comments:**

1. In Figure 1, the schematic illustrates entrainment from the cloud base. Does it mean that there is no lateral and cloud-top entrainment? Then what contributes to the hydrometeor evaporation at the cloud top?

2. The authors clearly explain that at higher aerosol loading, rather than being removed to the surface via rainout, it enhances aerosol regeneration. It is then transported to the free troposphere, where they remain available for reactivation and further aerosol-cloud interactions, which is clearly illustrated in Figure 1, which is a single-layered cloud. Now the question is, how does this relation in a multilayered cloud? So is this finding only apply to single-layer clouds?

3. This study investigates the aerosol impacts on shallow maritime tropical clouds, specifically the influence of aerosol budget. Out of four aerosol types, Mineral dust and ammonium sulphate also can act as INP. From Figure 2, it is evident that the cloud's top height reached up to 7 Km. How does it influence the aerosol budget? Is this also considered by calculating the aerosol budget?

4. Under increased aerosol loadings, the cloud droplets are smaller, and it enhances the cloud albedo. In addition, the regenerated aerosol particle interacts with the radiation. How does aerosol regeneration influences total and effective radiative forcing?

5. It would be much appreciated if you could include the model domain details as well (for example, latitude and longitude) in the experimental setup.

6. In section 2.3, it is mentioned that "..updrafts weaker than 1 ms-1 are necessarily excluded from the analysis". The model domain is over the ocean, and the model is initialised using CAMP$^2$Ex observations. What was the observed range of updraft velocities in CAMP$^2$Ex? I expect for the marine clouds, the updraft between 0.01 to $1\,\mathrm{m\,s}^{-1}$ would be quite strong. Is this a simulated case with a high vertical velocity?

**Minor Comments:**

1. Page 2, l34; please correct heterogenous to heterogeneous.

2. Page 2, l36; Please change cloud top to cloud tops

3. Table 1; Please expand ERA-5

4. Table 1; Please expand AGL

5. Table 1; Please mention LEAF-3

6. Page 5, l94; "Aerosol particles were initialized in the unactivated aerosol category." Cloud, please elaborate on it; what are these unactivated aerosol categories? It would be easier for the readers to understand.

7. Section 2.3: A very brief explanation of the Tracking and Object-Based Analysis of Clouds (tobac) would be helpful for the readers.

8. Page 8, l146: Is QC means quality control? Then please abbreviate it before (P7, l132).

9. Figure 2: It would be better to have alternative altitude levels (0,2,6,..) for better understanding. Also, please move a bit down the x and y-axis texts.

10. Figure 2 Caption: Please change g kg-1 to $g\,kg^{-1}$

11. Figure 2 Caption: Please add unit to potential temperature (K).

12. Figure 3: In Figure 3b, At lower initial surface concentration, except ammonium sulfate, all other aerosols increase with initial aerosol mass. An explanation would be useful for the readers.

13. Figure 4: Please add units to the parameters in the figure caption.

14. Figure 5: Please add units to the parameters in the figure caption and to the Figure titles.

15. Saturation effect for the rainout: Would it be possible to show this, perhaps an additional figure?

---

## Author Comment (AC1)

Leung et al. provide an interesting perspective on aerosol-cloud interactions in warm marine clouds by arguing that aerosol loading not only perturbs the clouds themselves but also the overall aerosol budget. These changes to the aerosol budget are primarily driven by changes to entrainment/detrainment and rainout. Overall I found the paper to be interesting, with nice figures, and it is a good fit for ACP. However, I would like to see more explanation of the experimental setup and analysis of the time-evolution of quantities the before recommending acceptance. Detailed comments below:

We thank the reviewer for their constructive and insightful suggestions, which have greatly improved the quality of our manuscript. Responses are indicated below in blue.

L44-45: it would be good to acknowledge here that aerosol changes can also produce changes in atmospheric circulation, which generate global-scale impacts and impacts on different cloud regimes. For example: Dagan (2022, JAMES) and Williams et al. (2022, Nature Climate Change).

We have included a sentence on this in lines 49-52: "Perturbations to the aerosol environment can also drive changes in the atmospheric circulation, with local, regional, and global impacts on cloud regimes (van den Heever et al., 2011; Haywood et al., 2013; Grant and van den Heever, 2014; Kim et al., 2016; Herbert et al., 2021; Williams et al., 2022; Dagan, 2022; Park and van den Heever, 2022; Leung and van den Heever, 2023).

L50: It took a few tries for me to understand this part of the sentence, could you please reword? "Aerosol-induced changes to clouds may feed back to how clouds and precipitation influence the aerosol field...". Maybe "Aerosol may alter the relationship between clouds and precipitation and the overall aerosol field..."?

We have altered this sentence in lines 54-55: "Aerosols may induce changes to clouds and precipitation, however these changes to clouds and precipitation may, in turn, influence the aerosol field, […]".

L76: I appreciate that you don't wish to repeat everything about these simulations, but a few more details would be helpful here. For example, do you include a diurnal cycle or does the "diurnal cycle" of Line 79 just refer to a 24hr period? By the sounds of it you included a diurnal cycle in the solar insolation, which I imagine would also alter the aerosol budget through changes in cloudiness? If indeed there is a diurnal cycle in the simulations it would be good to analyse whether these effects vary depending on the time of day (or at least argue why you *don't* do this).

We have now clarified that insolation does vary with the diurnal cycle in Table 1 and line 92: "[…], though the solar insolation varied according to the diurnal cycle".

While the reviewer makes a good suggestion to analyze the diurnal variation in these trends and we agree that such analysis would be interesting, we have decided not to include this in the paper. Given that the simulations were only run out for 48 hours, there are only two diurnal cycles to average over (and less if the first 7-8 hours of spin-up time are excluded). As such, we

do not feel confident in presenting any diurnal trends without running the simulations out for a longer time period.

That being said, we did some brief analysis of how the trends presented in this paper vary as a function of time of day, as shown in the figure below. Due to the aforementioned small number of full diurnal cycles, the trends are fairly noisy. However, there is generally a peak in clouds, precipitation, and associated aerosol regeneration and rain-out in the evening hours (centered on ~6PM). There are not clear trends in the diurnal cycle as a function of aerosol type or concentration. Overall, the trends we discuss in the paper (e.g., more regenerated aerosol with increasing aerosol loading) are fairly consistent throughout the diurnal cycle. Again though, we emphasize that given the number of diurnal cycles represented in these simulations, we choose not to include this analysis in the paper, though such analysis would certainly be interesting to do in the future with longer-term simulations.

[Figure]

**Figure R1.** Diurnal cycle of (a-d) number of raining clouds, (e-h) number of non-raining clouds, (i-l) mean rain rate, (m-p) percent regenerated aerosol mass, and (q-t) percent rained-out aerosol mass as a function of aerosol type and concentration. Values were averaged according to time of day in 2-hour bins. The first 8 hours of simulation time were excluded due to the low number of clouds. The aerosol budget terms show the change in each term over the 5-minute interval (as opposed to the accumulated terms shown in the main text of the paper) prior to diurnally averaging.

L82: "After initialization, the model was allowed to evolve freely without additional large-scale forcing." Without imposing large-scale forcing, how do you prevent the formation of deep convection in your domain? The lack of a large-scale forcing is confusing to me, why do the authors not just use an established case study like RICO?

In this study, which was done as part of the CAMP$^2$Ex campaign, we were specifically interested in cloud fields in the region around the Philippines, which is why we did not use a case study from other regions as the reviewer suggests. Certainly, similar analysis as we performed here could also be performed for other case studies and variability in trends across regions can be explored; we think that those results would be highly interesting, though would not be under the scope of this particular paper.

We used actual dropsonde observations to initialize the domain, since we are interested in understanding the formation and response of shallow clouds to the observed environments, as opposed to an idealized scenario with prescribed standard forcing. In our simulations, the environment develops as expected and forms clouds that are representative of those observed in reality; no deep convection develops, which is not surprising given that the dropsonde profile was specifically selected from a day without deep convection, i.e., it represents an environment that supports congestus clouds, but not deep convection.

Given that the type of cloud field we are simulating here is not dependent on large-scale subsidence for its formation, we have opted for this initialization from an observed profile to allow the environment and the clouds forming within it to evolve together, rather than prescribing a large-scale forcing which would force clouds back to a base state. We are of the opinion that our approach and the large-scale forcing approach suggested by the reviewer are equally valid, but our approach is more appropriate for the specific goals of this paper.

L90: Do the 'microphysics-radiation' experiments this include the Twomey effect for ice as well as liquid?

Yes, and we now explicitly state this in lines 100-101: "Both aerosol-radiation and microphysics-radiation interactions (liquid and ice phase) were included in the simulation."

Figure 1: Very nice schematic, I found it helpful!

Thank you!

L109: A bit pedantic, but I'm not sure if the use of "ensemble" is justified here (or indeed anywhere in the paper) as you only use one set of initial conditions per simulation. Instead, maybe just "multiple simulations were conducted where we varied X and Y...".

Yes, good point. We have replaced "ensemble" here and elsewhere in the paper with "set of simulations" or "simulations" as appropriate.

L120: SSA and other quantities have to defined at a specific wavelength, eg 550nm. Which is it? Also, for the ARI experiments it would be nice to also quote the AOD if the data is available, to more easily compare with other studies with simpler aerosol schemes.

RAMS, like many other cloud-resolving models, computes aerosol-radiation interactions integrated across multiple radiation bands rather than at each wavelength in order to save on computational cost (Saleeby and van den Heever, 2013). The quantities given in Table 2 are computed over the band covering 245-700nm, with a band midpoint of 472.5 nm, which we have now included in the table. We have also included information on AOD for the same radiation band in the supplementary information (Figure S1i-l).

L149: "Qualitatively similar cloud fields develop in all sixteen simulations..." It would be helpful if you could demonstrate this to the reader. For example, a figure showing the time-evolution of domain averaged fields would be helpful to get a sense of the variability over the 48hr period, and perhaps some sense of the spread in domain-avg properties across the experiments too.

This is a good suggestion. We have now added timeseries in the supplementary information (Figure S1).

L152: Again, I'm a bit confused how you don't get more deep convection if you don't impose large-scale temp/moisture tendencies?

Please see our above response to L82.

L162-164: Is this just a snapshot at the end of the simulation? How much variability is there in the timeseries? I'm left wondering exactly how representative these changes are.

The reviewer is correct that these are snapshots at the end of the simulation. However, we have found that the major trends in the percent aerosol mass that is regenerated or rained-out are consistent with time. We have added a sentence to this effect in lines 239-240: "While Figure 3 shows a snapshot in time, these trends are largely consistent throughout the course of the simulation (full timeseries shown in **Figure S2**)."

L169: Would the aerosol number not also be conserved? It sounds like if you are just tracking how this is partitioned across the four categories then you could also get a closed budget for aerosol number?

As it is treated in RAMS, aerosol number is not conserved when multiple aerosol particles are collected by one hydrometeor. We have added clarification of this on lines 201-203: "whereas all aerosol number is not conserved when multiple aerosol particles are collected by a single hydrometeor (i.e., it is assumed that interstitial aerosol that are collected by a droplet cohere with the original activated particle)."

L172: Would it be possible to show this, at least in the reply? 5% is actually quite large relative to the magnitude of these changes with aerosol loading in Fig. 3. Also, I'm confused about why

aerosol mass would be "lost" (i.e. not accounted for in your budget) due to dry deposition? Sorry if I'm missing something here.

Thank for your raising this point. We have now included the residual term in the supplementary material in Figure S2. Under the RAMS aerosol budget, only aerosol removal to the surface via wet deposition is tracked, and there is no budget term for dry deposition. We have now included a brief statement on this in lines 205-206: "treated as a residual that is lost due to dry deposition (which is represented in RAMS, but not tracked in the RAMS aerosol budget)". That being said, we emphasize that this residual term is quite small relative to all the other terms in the aerosol budget.

Figure 3: Please add a '100' marker to the x axis :)

We have now done this for Figures 3, 4, 5, 6, 7.

L179: 'simulations' not 'ensemble members'

We have replaced "ensemble members" here and elsewhere in the paper with "simulations"

L196: It is indeed clear from the figures, but I'm still wondering how representative these changes are if they're just calculated from snapshots? I assume there must be a decent bit of temporal variability in the simulated budget quantities?

As stated above, the snapshot at the end of the simulation appears representative of the overall trend, and we now discuss this lines 239-240: "While Figure 3 shows a snapshot in time, these trends are largely consistent throughout the course of the simulation (full timeseries shown in **Figure S2**)."

L236: Why would larger droplets have a smaller surface area? I find the wording confusing.

We meant here that the larger droplets had a proportionally smaller surface area compared to smaller droplets, but we understand that the wording was previously confusing. We have now rephrased this to say "a lower surface area-to-volume ratio" (lines 280-281).

L241: Regarding precipitation efficiency it would also be good to cite Lutsko et al. (2022, AGU Monographs) and Li et al. (2022, Nature Climate Change). Also it would be nice to discuss whether these results are consistent with the recent study by Dagan (2022, ACP) who also touched on changes in precipitation efficiency with aerosol loading.

Thank you for this recommendation. We have added citations to Lutsko et al. (2022) and Li et al. (2022) in line 288. While Dagan (2022) primarily focused on precipitation efficiency under different $CO_2$ concentrations in conjunction with different aerosol loadings, they found generally that increasing the aerosol concentration increases the precipitation efficiency—which is not the same as what we have found here (where precipitation efficiency changes non-monotonically with aerosol loading, but generally decreases). This discrepancy may be due to a myriad of factors, including different aerosol loadings or meteorological conditions tested, and may also be

due to the differences in grid spacing between the Dagan (2022) paper and this paper, where the 1km grid spacing in Dagan (2022) may not sufficiently resolve shallow cumulus clouds. We have added a sentence to the paper discussing this (lines 374-378): "Furthermore, this shows that the impact of increasing aerosol on precipitation efficiency is dependent on cloud type. This may explain differences between this study and recent studies such as Dagan (2022), which saw monotonic increases in precipitation efficiency with increasing aerosol. The latter study used a coarser grid spacing that was not resolving the shallow cumulus cloud field which we find in this work to have decreasing precipitation efficiency with increasing aerosol."

Section 4: Just wanted to say that I think this section is great!

Thank you very much for your comments!

Figure 10: Could you add another row above this which shows the baselines radiative cooling rates for each aerosol type? It's difficult to interpret just the changes alone.

This is a great idea! We have added panel (a) to show the baseline radiative heating rate for each aerosol type. The difference in heating rate between the different aerosol types is very small, and it is primarily the changes with increasing aerosol loading that are different as a function of aerosol type.

References:

Dagan, 2022 JAMES: https://agupubs.onlinelibrary.wiley.com/doi/full/10.1029/2022MS003368

Williams et al, 2022: https://www.nature.com/articles/s41558-022-01415-4

Lutsko et al 2022; https://www.authorea.com/doi/full/10.1002/essoar.10507822.1

Li et al, 2022: https://www.nature.com/articles/s41558-022-01400-x

Dagan 2022 ACP; https://acp.copernicus.org/articles/22/15767/2022/

---

## Author Comment (AC2)

The authors have investigated aerosol-induced changes to cloud properties in shallow maritime tropical clouds. The modelling study demonstrates that increasing the aerosol loading enhances aerosol regeneration at the expense of rainout regardless of aerosol type. Such studies are very important for a process-based understanding of aerosol-cloud interactions. However, clarity and precision are lacking in the manuscript. However, clarity and precision are lacking in the manuscript.

The topic "Aerosol-cloud impacts on aerosol detrainment and rainout in shallow maritime tropical clouds" is of interest and fits the scope of EGUsphere. The paper can be accepted with major revisions. Please see the comments below.

We thank the reviewer for their very helpful comments, which have greatly improved the quality of our manuscripts. Responses are listed below in blue.

Major Comments:
1. In Figure 1, the schematic illustrates entrainment from the cloud base. Does it mean that there is no lateral and cloud-top entrainment? Then what contributes to the hydrometeor evaporation at the cloud top?
Thank you for raising this point. We initially chose to illustrate entrainment at the cloud base only in order to simplify the figure. However, the reviewer is correct that there is also lateral and cloud top entrainment, which we have now included in the revised Figure 1.

2. The authors clearly explain that at higher aerosol loading, rather than being removed to the surface via rainout, it enhances aerosol regeneration. It is then transported to the free troposphere, where they remain available for reactivation and further aerosol-cloud interactions, which is clearly illustrated in Figure 1, which is a single-layered cloud. Now the question is, how does this relation in a multilayered cloud? So is this finding only apply to single-layer clouds?
The cloud field we simulated in this study was composed only of single-layer cumulus and congestus clouds, the kind of which is illustrated in Figure 1. However, the reviewer raises an interesting point about multilayered clouds. We suspect that in such a scenario, there would be multiple detrained aerosol layers at different levels where clouds are detraining. We saw similar scenarios in the field during the CAMP²Ex field campaign, where congestus and deep convective clouds were simultaneously detraining. However, we would need to run further simulations with multilayered cloud fields to confirm this. Furthermore, aerosol that are detrained from lower clouds then have implications for entrainment by clouds at upper levels, as we now discuss in lines 469-471: "Instead, the aerosol particles are regenerated aloft, where they form an aerosol detrainment layer (or potentially, in the case of multilayer clouds, detrainment layers) that can serve as an aerosol source for future midlevel and multilayer clouds (Leung and van den Heever, 2022)."

3. This study investigates the aerosol impacts on shallow maritime tropical clouds, specifically the influence of aerosol budget. Out of four aerosol types, Mineral dust and ammonium sulphate also can act as INP. From Figure 2, it is evident that the cloud's top height reached up to 7 Km. How does it influence the aerosol budget? Is this also considered by calculating the aerosol budget?

Because the focus of this study was on warm phase convection (shallow cumulus and congestus clouds), we did not specifically vary the INP between simulations. Instead, in all simulations, we prescribed a fixed concentration if INP starting at 0.01 # $cm^{-3}$ at the surface and decaying exponentially in the vertical, with a scale height of 7km. We describe this in lines 99-100: " In all simulations, ice-nucleating particles (INP) were also initialized with concentrations starting at 0.01 # $cm^{-3}$ at the surface and the same vertical structure as the aerosol field." Additionally, aerosol mass could be transferred when liquid water is transferred to ice species; any aerosol mass existing within the ice species was also included in the "in-hydrometeor" term of the aerosol budget, as specified in lines 108-109: "As water mass was transferred between hydrometeor species (i.e., cloud, drizzle, rain, ice, snow, aggregates, hail, graupel), a corresponding fraction of aerosol was also transferred."

4. Under increased aerosol loadings, the cloud droplets are smaller, and it enhances the cloud albedo. In addition, the regenerated aerosol particle interacts with the radiation. How does aerosol regeneration influences total and effective radiative forcing?
We were not entirely sure of what the reviewer is asking here.

If the question is whether the regenerated aerosol particles are accounted for in the total and effective radiative forcing, then the answer is yes. Regenerated aerosol particles are assigned the same radiative properties as the unprocessed aerosol, and thus still contribute to the overall radiative forcing.

If the question is requesting more detailed information about the role of these regenerated aerosol particles of different types on the radiative budget, then while we agree this is an important consideration, this is beyond the consideration of this paper. However, we do show and discuss the radiative heating rates and differences as a result of aerosol loading and type in Section 5.

5. It would be much appreciated if you could include the model domain details as well (for example, latitude and longitude) in the experimental setup.
We have now included information about the domain latitude and longitude in Table 1.

6. In section 2.3, it is mentioned that "..updrafts weaker than 1 ms-1 are necessarily excluded from the analysis". The model domain is over the ocean, and the model is initialised using CAMP2Ex observations. What was the observed range of updraft velocities in CAMP2Ex? I expect for the marine clouds, the updraft between 0.01 to 1 m s−1 would be quite strong. Is this a simulated case with a high vertical velocity?
Thank you for raising this point. The simulated cloud field produced tens of thousands of updrafts with maximum vertical velocities above 1 m $s^{-1}$ (see Supplementary Figure 1e-h, which shows the maximum updraft is 10-30 m $s^{-1}$ for most of the simulation period). Although complete measurements of the updraft velocity PDFs during $CAMP^2Ex$ are unfortunately not as yet available, our simulations are consistent with observations of numerous individual updrafts over 1 m $s^{-1}$ were observed (Reid et al. 2023). The PDF of vertical velocity here is also in keeping with past modelling studies of the same region (e.g., Leung and van den Heever 2022, Sokolowsky et al. 2022).

As the reviewer says, there are indeed numerous updrafts between 0.01 to 1 m s$^{-1}$. However, these weak updrafts do not significantly impact the domain-wide precipitation or aerosol budget. The table below shows the percent of precipitation and aerosol rainout attributed to all updrafts above 1 m s$^{-1}$ and to features which pass all our QC thresholds. Weak updrafts below 1 m s$^{-1}$ contribute less than 1% of all precipitation and less than 2% of all aerosol rainout, and thus are not important for the results we present.

| | | All Updrafts > 1 m s$^{-1}$ | | Features passing QC | |
|---|---|---|---|---|---|
| | | Precipitation | Aerosol Rainout | Precipitation | Aerosol Rainout |
| Ammonium Sulphate | 100 cm$^{-1}$ | 99.2 | 98.9 | 81.7 | 83.0 |
| | 500 cm$^{-1}$ | 99.1 | 98.9 | 77.3 | 78.5 |
| | 1000 cm$^{-1}$ | 99.0 | 98.8 | 78.6 | 77.2 |
| | 1500 cm$^{-1}$ | 98.9 | 98.9 | 74.8 | 74.0 |
| Sea Salt | 100 cm$^{-1}$ | 99.2 | 98.9 | 82.3 | 85.7 |
| | 500 cm$^{-1}$ | 99.1 | 99.0 | 91.1 | 81.6 |
| | 1000 cm$^{-1}$ | 99.0 | 98.8 | 78.4 | 77.2 |
| | 1500 cm$^{-1}$ | 99.1 | 99.1 | 76.3 | 76.3 |
| Absorbing Carbon | 100 cm$^{-1}$ | 99.1 | 98.0 | 79.2 | 82.2 |
| | 500 cm$^{-1}$ | 99.0 | 98.3 | 80.0 | 80.4 |
| | 1000 cm$^{-1}$ | 99.2 | 99.0 | 79.6 | 78.6 |
| | 1500 cm$^{-1}$ | 99.1 | 98.9 | 75.4 | 74.4 |
| Mineral Dust | 100 cm$^{-1}$ | 99.1 | 97.9 | 81.2 | 82.3 |
| | 500 cm$^{-1}$ | 99.1 | 98.4 | 79.3 | 81.3 |
| | 1000 cm$^{-1}$ | 99.1 | 98.6 | 78.2 | 78.4 |
| | 1500 cm$^{-1}$ | 99.2 | 98.9 | 76.0 | 75.4 |

We have added a statement on this in lines 170-174: "Although there are numerous updrafts with maximum vertical velocities below 1 m s$^{-1}$ present in the simulation, we found that such weak updrafts do not contribute significantly to the precipitation or aerosol budget, accounting for less than 2% of precipitation and aerosol rainout. Furthermore, after applying all our QC thresholds, we found that a vast majority of the falling rain and rained out aerosol (75-80%) could be attributed to the remaining features, and that this was consistent across all these simulations."

Minor Comments:
1. Page 2, l34; please correct heterogenous to heterogeneous.
We have changed this.

2. Page 2, l36; Please change cloud top to cloud tops
We have changed this.

3. Table 1; Please expand ERA-5
We have now defined ERA-5 on page 4, line 81.

4. Table 1; Please expand AGL
We have now defined AGL

5. Table 1; Please mention LEAF-3

We have now defined LEAF-3.

6. Page 5, l94; "Aerosol particles were initialized in the unactivated aerosol category." Cloud, please elaborate on it; what are these unactivated aerosol categories? It would be easier for the readers to understand.

We have updated this sentence (lines 104-106) to better explain this: "Upon initialization, all aerosol particles are initially categorized as unactivated aerosol, i.e., aerosol particles which have not yet been activated in cloud droplets."

7. Section 2.3: A very brief explanation of the Tracking and Object-Based Analysis of Clouds (tobac) would be helpful for the readers.

The first paragraph in this section does provide a description of how *tobac* works (feature identification, tracking, and segmentation).

8. Page 8, l146: Is QC means quality control? Then please abbreviate it before (P7, l132).

We have now defined QC on line 155.

9. Figure 2: It would be better to have alternative altitude levels (0,2,6,..) for better understanding. Also, please move a bit down the x and y-axis texts.

Thank you for this suggestion. We have changed the tick marks for the altitude labels and moved down the x- and y-axis labels.

10. Figure 2 Caption: Please change g kg-1 to g $kg^{-1}$

We have now changed this.

11. Figure 2 Caption: Please add unit to potential temperature (K).

We have now added this.

12. Figure 3: In Figure 3b, At lower initial surface concentration, except ammonium sulfate, all other aerosols increase with initial aerosol mass. An explanation would be useful for the readers.

The trend that the reviewer is pointing to between the 100 and 500 $\#$ $cm^{-3}$ simulations is actually not consistent in time (see Supplementary Figure 2e-h). Given that any apparent trends in the percent of aerosol in the in-hydrometeor category are small in magnitude, not stable in time, not consistent between aerosol types, and not monotonic with concentration, the evidence does not appear to support a physically driven trend in this aerosol category. We have therefore chosen not to focus our discussion on this.

13. Figure 4: Please add units to the parameters in the figure caption.

We have added the units in the figure caption.

14. Figure 5: Please add units to the parameters in the figure caption and to the Figure titles.

We have added in the figure caption that all process efficiencies are in % of condensed vapor. This is already indicated on each panel of the figure.

15. Saturation effect for the rainout: Would it be possible to show this, perhaps an additional figure?

We were not entirely sure of what the reviewer is asking here. Figure 7c does show the drop-off of the aerosol rainout rate with increasing aerosol loading, which demonstrates the saturation effect we describe. It is possible that with further simulations at increased aerosol loadings, the graph could be extended to extend this trend further; however, this is beyond the scope of this current paper.

---

## Referee Report (RR1)

The authors significantly improved the manuscript, addressing almost all the points raised in the previous version. Still, I think further revision would benefit the readers. Therefore I recommend accepting with minor revisions. Minor comments are given below.

**Minor Comments:**

1. In Section 2.1, it is mentioned that the domain is located over the ocean, and the simulation was initialized using observation from CAMP2Ex. However, the information on the exact domain (latitude and longitude) is missing in the section ( Is this the domain (between 8-9°N, 119-120°E))?. Also, the authors haven't mentioned the days (Dates) of the simulation and the dominating weather pattern during the period of simulation. This information would help the readers understand the region's features and the weather patterns during the simulation.

2. Page 9, l80; Please change interstitial aerosol to interstitial aerosols

3. Page 11, l211; Please change timeseries to time series

4. Page 11, l217; Please change suggests to suggest

5. Page 14, l262-263; Please rephrase the sentence "That is to say, they showed that increased aerosol loading caused less efficient rain formation and therefore less efficient aerosol removal through rainout."

6. Page 21, l402; Please change feed back to feedback

7. Page 22, l412; Please add the surface

8. Page 22, l413; Please change have to has

9. Page 22, l432; Please change that is to that are

---

## Author Response (AR2)

**Reviewer 1:**
We thank the reviewer for their comment. Our response is indicated below in blue.

L205 of the revised manuscript reads "the trends in this category vary temporally (Figure S1)", but I think the sentence should be referencing Figure S2?
We have corrected this now. Line 205 now references Figure S2.

**Reviewer 2:**
The authors significantly improved the manuscript, addressing almost all the points raised in the previous version. Still, I think further revision would benefit the readers. Therefore I recommend accepting with minor revisions. Minor comments are given below.
We thank the reviewer for their comments. Our responses are indicated below in blue.

Minor Comments:
1. In Section 2.1, it is mentioned that the domain is located over the ocean, and the simulation was initialized using observation from CAMP2Ex. However, the information on the exact domain (latitude and longitude) is missing in the section (Is this the domain (between 8-9∘N, 119-120∘E))?. Also, the authors haven't mentioned the days (Dates) of the simulation and the dominating weather pattern during the period of simulation. This information would help the readers understand the region's features and the weather patterns during the simulation.
The domain is centered on 8.75ºN, 119.75ºE in the Sulu Sea, and covers a similar region to the ERA-5 data used for initialization. We have updated the text in Table 1 to include the center point and dates of the simulation. However, we want to clarify that because this is an idealized simulation and not a case study, we do not intend to replicate the exact weather patterns during a specific time period, and instead aim to simulate similar/realistic congestus clouds as were observed during CAMP$^2$Ex.

2. Page 9, l80; Please change interstitial aerosol to interstitial aerosols
Line 180 now reads "interstitial aerosol particles".

3. Page 11, l211; Please change timeseries to time series
Done!

4. Page 11, l217; Please change suggests to suggest
Done!

5. Page 14, l262-263; Please rephrase the sentence "That is to say, they showed that increased aerosol loading caused less efficient rain formation and therefore less efficient aerosol removal through rainout."
We are not sure what the reviewer is asking us to change about this sentence, and so we have retained it as is.

6. Page 21, l402; Please change feed back to feedback
For clarity, we have changed this line to read: "how aerosol impacts on shallow maritime tropical clouds (cumulus, congestus) subsequently modify the aerosol budget"

7. Page 22, l412; Please add the surface

We are not sure where in this line the reviewer is suggesting we make a change, so we have retained line 412 as is.

8. Page 22, l413; Please change have to has

For clarity, line 416 now reads: "each droplet that does form through stochastic collisions with other droplets then has more cloud water available for collection"

9. Page 22, l432; Please change that is to that are

Line 432 now reads: "[…] found that a mixed domain-wide trend in rain amounts is driven by […]".